# Development of a Serum Proteomic-Based Diagnostic Model for Lung Cancer Using Machine Learning Algorithms and Unveiling the Role of SLC16A4 in Tumor Progression and Immune Response

**DOI:** 10.3390/biom15081081

**Published:** 2025-07-26

**Authors:** Hanqin Hu, Jiaxin Zhang, Lisha Zhang, Tiancan Li, Miaomiao Li, Jianxiang Li, Jin Wang

**Affiliations:** School of Public Health, Suzhou Medical College of Soochow University, Suzhou 215123, China

**Keywords:** lung cancer, diagnostic model, serum proteomics, SLC16A4, machine learning

## Abstract

Early diagnosis of lung cancer is crucial for improving patient prognosis. In this study, we developed a diagnostic model for lung cancer based on serum proteomic data from the GSE168198 dataset using four machine learning algorithms (nnet, glmnet, svm, and XGBoost). The model’s performance was validated on datasets that included normal controls, disease controls, and lung cancer data containing both. Furthermore, the model’s diagnostic capability was further validated on an independent external dataset. Our analysis identified SLC16A4 as a key protein in the model, which was significantly downregulated in lung cancer serum samples compared to normal controls. The expression of SLC16A4 was closely associated with clinical pathological features such as gender, tumor stage, lymph node metastasis, and smoking history. Functional assays revealed that overexpression of SLC16A4 significantly inhibited lung cancer cell proliferation and induced cellular senescence, suggesting its potential role in lung cancer development. Additionally, correlation analyses showed that *SLC16A4* expression was linked to immune cell infiltration and the expression of immune checkpoint genes, indicating its potential involvement in immune escape mechanisms. Based on multi-omics data from the TCGA database, we further discovered that the low expression of *SLC16A4* in lung cancer may be regulated by DNA copy number variations and DNA methylation. In conclusion, this study not only established an efficient diagnostic model for lung cancer but also identified SLC16A4 as a promising biomarker with potential applications in early diagnosis and immunotherapy.

## 1. Introduction

Lung cancer remains one of the most prevalent and deadly cancers worldwide, with a significant impact on global health [1,2]. Despite advancements in early detection and treatment strategies, the prognosis for patients diagnosed with lung cancer is often poor due to its late-stage diagnosis [3]. The conventional diagnostic methods, such as imaging and biopsy, although valuable, are often invasive, time-consuming, and limited by subjective interpretation [4]. As a result, there is an increasing demand for non-invasive, efficient, and reliable diagnostic approaches.

In recent years, proteomics, particularly serum proteomics, has emerged as a promising tool in the early detection of various cancers, including lung cancer [5,6]. Serum proteomic profiling offers valuable insights into the molecular alterations associated with tumor development, providing a potential biomarker for cancer diagnosis [7]. However, the complexity and high-dimensional nature of proteomic data present significant challenges in the accurate identification of biomarkers and the development of robust diagnostic models.

Machine learning (ML) algorithms have gained substantial attention in medical research due to their ability to handle large and complex datasets, extract meaningful patterns, and make predictions [8,9,10]. The application of machine learning to proteomic data has shown great promise in the development of diagnostic models for cancer detection. Several studies have utilized various machine learning algorithms, including artificial neural networks (ANNs), support vector machines (SVM), gradient boosting machines (GBM), and regularized regression models, such as elastic net (GLMNET), to develop predictive models for cancer diagnosis [11,12,13].

This study aims to develop and evaluate diagnostic models for lung cancer detection using serum proteomic data. Specifically, we apply four machine learning algorithms—NNET, GLMNET, XGBoost, and SVM—to build classification models that can differentiate between lung cancer patients and healthy controls. By providing a comprehensive analysis of model performance, this study contributes to the growing body of knowledge on machine learning applications in cancer diagnostics and offers insights into the potential clinical implementation of these methods for improving early detection and patient outcomes.

## 2. Materials and Methods

### 2.1. Data Source

The protein microarray data and clinical information for GSE168198 were obtained from the Gene Expression Omnibus (GEO). The study was divided into two phases.

Phase I: Training and Internal Testing Sets

In Phase I, we utilized the BCBIO HUMAN CUSTOM PROTEOME MICROARRAY platform, annotated with GPL29810, which can detect 89 proteins. From the GSE168198 dataset, a total of 180 serum samples analyzed using this platform were used. These samples were divided into a training set and an internal testing set in a 7:3 ratio, resulting in 144 samples for training and 36 samples for internal testing. This division was carried out to ensure that the model could be rigorously trained and internally validated before moving to external validation.

Phase II: External Validation Set

Phase II involved the CDI LABS HUMAN PROTEOME MICROARRAY v4.0 platform, annotated with GPL29809, which is capable of detecting 16,048 proteins. The GSE168198 dataset included 16 serum samples analyzed using this platform. These samples were used exclusively as an external validation set to assess the model’s performance on an independent dataset, thereby confirming its generalizability and robustness.

Data for lung adenocarcinoma (LUAD) and lung squamous cell carcinoma (LUSC), along with clinical information, were extracted from The Cancer Genome Atlas (TCGA) database. After merging, the lung cancer dataset contained 1122 samples, including 109 normal tissue samples and 1013 lung cancer tissue samples.

### 2.2. Differential Protein Expression Analysis

In this study, differential expression analysis of plasma proteomics data was performed using the limma package in R (version 4.3.1) to identify significantly differentially expressed plasma proteins between the cancer and control groups. First, the expression values were log-transformed (log2(x + 1)) to reduce data skewness. Then, the data were filtered based on disease type (“LungCancer” and “NegativeControl”), retaining only relevant samples. Differential analysis was conducted using the limma package, and the results were extracted. Significance was determined based on a set threshold (logFC > 0.3 and adjust *p* value < 0.05). Upregulated and downregulated proteins were subsequently extracted to form the feature set required for building subsequent machine learning models.

### 2.3. Machine Learning Model Construction

Based on the results of differential expression analysis, a series of machine learning models were constructed to classify cancer and control groups. First, the dataset obtained from differential expression analysis was divided into training and testing sets, with 70% used for training and 30% for testing, ensuring that the training set contained diagnostic labels. The target variable was converted to a factor type to accommodate the machine learning models. Both the training and testing sets were standardized, including centering and scaling, to eliminate differences in measurement units across features.

During the model training phase, the “train” function from the “caret” package [14] was used, and 10-fold cross-validation was employed to evaluate model performance. Several machine learning algorithms were applied, including Elastic Net, Support Vector Machine (SVM), XGBoost, and Neural Networks, with hyperparameter tuning for each model to optimize classification performance. Additionally, SMOTE (Synthetic Minority Over-sampling Technique) was used to address the class imbalance issue, ensuring that the model could better learn the features of minority class samples during training [15]. Finally, a custom evaluation function was used to assess the model’s performance, including calculating metrics such as the Area Under the ROC Curve (AUC), accuracy, sensitivity, specificity, precision, and F1-score, to select the best model for subsequent analysis.

### 2.4. Model Validation

To evaluate the performance of the constructed models, validation was conducted on datasets containing different control samples, including disease controls, healthy controls, and mixed controls. First, uniform preprocessing was applied to the three datasets and the external validation set to ensure data standardization and comparability. The models, constructed based on the training set, were then applied to these datasets. During the evaluation process, model performance was primarily measured using multiple metrics, including accuracy, sensitivity, specificity, precision, F1-score, and AUC (Area Under the Curve).

### 2.5. Key Protein Selection

Based on model validation, this study further conducted the selection of key proteins associated with lung cancer diagnosis. Through feature importance analysis in the machine learning models, we identified several key proteins with potential roles in disease diagnosis. We evaluated the contribution of each protein across different models and extracted the top features from each model using the variable importance assessment (“varImp” function). Key proteins consistently identified across multiple models demonstrated high discriminative power for lung cancer diagnosis. By further analyzing the consistency of these proteins across models and visualizing the results with a Venn diagram, we identified two proteins that were common across all four models.

### 2.6. Expression Analysis of SLC16A4

High-throughput sequencing data from the TCGA database were used, including TPM-normalized data for lung adenocarcinoma (LUAD) and lung squamous cell carcinoma (LUSC), to analyze the expression differences between tumor and normal samples. The expression differences of the *SLC16A4* gene in tumor and normal tissues from the LUAD and LUSC cohorts were analyzed using the PCAS tool, which assesses RNA and protein expression levels [16]. Additionally, the GCAS online tool was employed to analyze the expression of this gene across multiple lung cancer datasets in the GEO database.

### 2.7. Correlation Analysis

To investigate the biological and clinical significance of *SLC16A4* in lung cancer, we analyzed the correlations between risk scores and the expression of oncogenes, tumor mutational burden (TMB), microsatellite instability (MSI), tumor stemness, immune regulatory gene expression, immune cell infiltration, and drug sensitivity scores. Oncogene data were obtained from the ONGene database (http://www.ongene.bioinfo-minzhao.org, accessed on 17 March 2025) [17]. A total of 11 immune checkpoint genes (ICGs) were extracted from previous studies [18]. Immune cell infiltration scores were sourced from the TIMER2.0 database (http://timer.cistrome.org/, accessed on 17 March 2025) [19]. Drug sensitivity was evaluated using data from the Cancer Drug Sensitivity Genomic Database (GDSC) through the “oncoPredict” tool [20]. Correlation analyses were performed using the Spearman method (the “psych” package).

### 2.8. Enrichment Analysis

Gene set enrichment analysis (GSEA) was employed to investigate the pathways and biological functions associated with *SLC16A4*. Initially, all genes correlated with *SLC16A4* expression were identified through correlation analysis and ranked based on their correlation coefficients. GSEA was performed using the “ClusterProfiler” R package, which utilizes predefined gene sets, including C2 (c2.cp.kegg.v2023.2.entrez.gmt) and C5 (c5.go.bp.v2023.2.entrez.gmt) [21]. Pathways significantly associated with risk scores were identified based on the normalized enrichment score (NES) and false discovery rate (FDR).

### 2.9. Cell Culture

The non-small cell lung cancer (NSCLC) cell lines used in this study included A549 and H1299, both of which were purchased from the American Type Culture Collection (ATCC) and cultured under standard conditions. A549 and H1299 cells were cultured in RPMI-1640 medium supplemented with 10% fetal bovine serum (FBS). Cells were maintained in a humidified incubator at 37 °C with 5% CO_2_. Passaging was performed when the cells reached 70–80% confluence. Prior to use, morphological inspection confirmed that all cell lines were free of contamination.

### 2.10. Overexpression Plasmid Construction

To investigate the function of the *SLC16A4* gene, we constructed an *SLC16A4* overexpression plasmid. First, specific primers were designed, and the coding region of the human *SLC16A4* gene was amplified by PCR. The PCR products and pCDH expression vector were then digested using the restriction enzymes XbaI and KpnI, and the *SLC16A4* gene fragment was ligated into the vector. The ligation mixture was transformed into E. coli DH5α, and positive clones were selected and verified by sequencing to confirm correct plasmid construction. Finally, plasmids were purified using the Plasmid Mini Kit to provide high-quality *SLC16A4* overexpression plasmids for subsequent cell transfection experiments.

### 2.11. CCK8 Cell Proliferation Assay

Cell proliferation was assessed using the Cell Counting Kit-8 (CCK-8) assay (Beyotime, Nantong, China). The experimental procedure was as follows: Transfected A549 or H1299 cells were seeded in a 96-well plate at a density of 2 × 10^3^ cells per well. After 24 h of incubation, different concentrations of drugs or medium were added. At 24, 48, and 72 h post-treatment, 10 μL of CCK-8 solution was added to each well, and the cells were incubated for an additional 2 h. The optical density (OD) at 450 nm was measured using a microplate reader (BioTek, Winooski, VT, USA). The cell proliferation inhibition rate was calculated based on the OD values, and proliferation curves were generated for different time points and drug concentrations.

### 2.12. EdU Cell Proliferation Assay

Cell proliferation was assessed using the 5-ethynyl-2′-deoxyuridine (EdU) incorporation assay. Transfected A549 or H1299 cells were seeded into 96-well plates and treated with the respective compounds. Following treatment, cells were incubated with EdU labeling solution for 2 h. After fixation, staining was performed according to the manufacturer’s protocol. EdU-positive cells were visualized and counted under a fluorescence microscope, and the proliferation index was calculated.

### 2.13. β-Galactosidase Staining for Cellular Senescence

To evaluate cellular senescence, β-galactosidase staining was performed using a commercially available senescence-associated β-galactosidase assay kit. Transfected cells were seeded into culture dishes and treated with the indicated drugs or stimuli. Cells were then stained following the manufacturer’s instructions. Senescent cells, identified by blue staining under light microscopy, were quantified. The proportion of senescent cells was calculated and compared across experimental groups.

### 2.14. Quantitative PCR (qPCR)

Quantitative PCR (qPCR) was used to determine the relative expression levels of CDKN1A and CDKN2A, with ACTB serving as an internal control. Total RNA was extracted and reverse-transcribed into cDNA. Each qPCR reaction was prepared using SYBR Green Master Mix, gene-specific primers (for CDKN1A, CDKN2A, and ACTB), and cDNA templates. The amplification protocol consisted of an initial denaturation at 95 °C for 30 s, followed by 40 cycles of 95 °C for 5 s and 60 °C for 30 s. Relative gene expression was calculated using the 2^−ΔΔCt^ method, where ΔCt represents the difference between the Ct values of the target gene and ACTB, and ΔΔCt indicates the difference between experimental and control groups.

### 2.15. Statistical Analysis

Statistical analyses were performed using GraphPad Prism version 8.3.0 (GraphPad Software, LLC, Boston, MA, USA). Data are presented as mean ± standard deviation (SD). Differences between two groups, such as gene expression levels between tumor and normal samples, gene expression differences in 5-aza treated vs. NC groups in DNA methylation validation experiments, gene expression differences between SLC16A4 overexpression group and empty vector transfection group, and gene expression differences between clinical pathological features with only two categories, were assessed using an unpaired Student’s *t*-test. Comparisons among multiple groups, such as gene expression differences across clinical pathological features with more than two categories, were conducted using one-way analysis of variance (ANOVA). All statistical tests were two-sided, and *p* < 0.05 was considered statistically significant.

## 3. Results

### 3.1. Dataset

Serum proteomic data and corresponding clinical data from lung cancer patients, healthy individuals, and other disease control volunteers were obtained from the GEO database (GSE168198). The basic demographic and clinical data for the training set, internal testing set, and validation set are summarized in Table 1. These data were carefully reviewed to ensure no significant differences in the major clinicopathological characteristics between the training set, testing set, and the entire TCGA-LUAD dataset, as expected (Table 1). This comparison confirms the representativeness and reliability of our datasets for lung cancer diagnosis modeling.

### 3.2. Model Construction

We analyzed the differential expression of plasma proteins in lung cancer patients using the GSE168198 dataset, identifying 23 proteins significantly altered between lung cancer patients and negative controls (Figure 1A, Appendix A). Using these proteins, we constructed diagnostic models based on four machine learning algorithms: nnet, glmnet, sgboost, and svm. The performance of these models was assessed through Receiver Operating Characteristic (ROC) curves.

As shown in Figure 1B and Figure 2, the XGBoost model exhibited the highest diagnostic performance with an AUC of 0.985, followed by nnet with an AUC of 0.970. The svm and glmnet models achieved AUC values of 0.897 and 0.861, respectively. Additionally, the XGBoost model demonstrated the highest accuracy in the training dataset, while the svm model achieved the highest accuracy in the testing dataset (Figure 2). The nnet model showed the highest sensitivity in both datasets, and the XGBoost model had the highest specificity in the training dataset (Figure 2). We further validated these models on an internal testing set, and the ROC curves as shown in Figure 2. The nnet model maintained an AUC of 0.886, while glmnet, XGBoost, and svm models achieved AUC values of 0.827, 0.840, and 0.830, respectively. After comparing our findings with previously reported tumor markers, including MYC, TP53, and HER2, our study further demonstrates the superiority of the machine learning model for the early diagnosis of lung cancer (Appendix A).

We analyzed the predictive performance of four machine learning models (XGBoost, glmnet, nnet, and svm) across different control datasets (Figure 3). The results revealed that the models performed best on the negative control dataset, with nnet and XGBoost achieving the highest AUC values of 0.923 and 0.918, respectively. In contrast, when tested on datasets containing only disease controls, the AUC values for these two algorithms were lower, at 0.756 and 0.774, respectively. When both negative controls and disease controls were included, the AUC values for nnet and XGBoost improved to 0.839 and 0.846, respectively.

### 3.3. Model Validation

We further evaluated the performance of the four diagnostic models on a small external cohort consisting of 8 healthy individuals and 8 lung cancer patients (Figure 4A–D). The results demonstrate that all models performed well on the external dataset, with the svm algorithm showing the highest AUC of 0.922. XGBoost followed closely with an AUC of 0.906, while glmnet and nnet had AUC values of 0.875 and 0.867, respectively. These findings suggest that all four models exhibit strong predictive performance, with svm and XGBoost providing the most accurate classifications.

### 3.4. Downregulation of SLC16A4 in Lung Cancer

To identify key proteins associated with lung cancer diagnosis, we first analyzed the top 5 most important proteins from each of the four machine learning models. The intersection of these proteins is shown in Figure 5A,B and Appendix A, with SLC16A4 and XAGE1A identified as the common proteins across all models. Due to the upregulation of the candidate biomarker XAGE1A in lung cancer tissues, which is inconsistent with its downregulated expression in plasma (Appendix A), we initially chose another candidate biomarker SLC16A4 for further analysis.

As shown in Figure 6A,B, *SLC16A4* mRNA expression was significantly lower in lung cancer samples than in normal tissues in the TCGA LUAD and LUSC datasets. Similar findings were observed in the CPTAC LUAD and LUSC datasets, where mRNA expression of *SLC16A4* was consistently reduced in cancer tissues (Figure 6C,D). Furthermore, protein expression levels of SLC16A4 in the CPTAC LUAD and LUSC datasets were also significantly lower in lung cancer tissues compared to normal tissues (Figure 6E,F). In addition, analysis of multiple lung cancer datasets in the GEO database confirmed the consistent downregulation of *SLC16A4* mRNA in lung cancer tissues across various cohorts (Figure 6G). Finally, Kaplan–Meier survival analysis using the KMPlotter online tool indicated that low expression of *SLC16A4* was associated with poorer overall survival in lung cancer patients, highlighting its potential as a prognostic biomarker (Figure 6H).

### 3.5. SLC16A4 Is Regulated by Copy Number Variation and DNA Methylation

Furthermore, we investigated the potential mechanisms underlying the downregulation of *SLC16A4* in lung cancer using multi-omics data from TCGA. In both the TCGA lung adenocarcinoma and squamous cell carcinoma cohorts, *SLC16A4* exhibited frequent copy number deletions, particularly single-copy deletions (Figure 7A). Further analysis revealed a positive correlation between *SLC16A4* expression and copy number (r = 0.177, *p* < 0.001) (Figure 7B). Methylation regulatory analysis showed a negative correlation between *SLC16A4* expression and DNA methylation levels (Figure 7C), with the methylation level of the cg00961640 site in the promoter region showing the strongest negative correlation with gene expression (r = −0.657, *p* < 0.001) (Figure 7D). Further differential analysis revealed that the methylation levels at multiple sites were significantly elevated in the TCGA lung adenocarcinoma and squamous cell carcinoma datasets (Figure 7E). Upon treatment with the methylation inhibitor 5-aza to suppress DNA methylation, the expression of *SLC16A4* was significantly upregulated in the lung cancer cell lines H1299 and A549 (Figure 7F). These findings suggest that *SLC16A4* expression is regulated by both copy number gain and promoter region demethylation.

### 3.6. Association of SLC16A4 with Clinical Characteristics in Lung Cancer

Further analysis was performed to investigate the correlation between *SLC16A4* expression and clinical pathological features using the TCGA lung cancer dataset. After grouping based on different clinical characteristics, we found that *SLC16A4* expression was significantly associated with gender, tumor invasion depth (M), lymph node metastasis (N), distant metastasis (T), tumor stage, and smoking history in lung cancer patients (Table 2). Notably, *SLC16A4* expression in male lung cancer tissues was significantly lower than in female lung cancer tissues (*p* < 0.001, Table 2). Regarding tumor staging, *SLC16A4* expression was significantly lower in early-stage tumors (Stage I-III) compared to late-stage tumors (Stage IV) and normal tissues (Figure 8A). In relation to smoking history, lung cancer tissues from never-smokers and patients who had quit smoking for more than 15 years exhibited higher *SLC16A4* expression than those from current smokers and individuals who had quit smoking for less than 15 years (Figure 8B).

### 3.7. Association of SLC16A4 with Immune Cell Infiltration

We further analyzed the relationship between *SLC16A4* expression and immune cell infiltration in lung cancer. Figure 9A shows the correlation between *SLC16A4* expression and the infiltration of various immune and stromal cells, calculated using the XCELL algorithm in the TCGA lung cancer dataset. *SLC16A4* expression was significantly correlated with stromal score (r = 0.525, Figure 9B), microenvironment score (r = 0.324, Figure 9C), and cancer-associated fibroblast (CAF) infiltration (r = 0.424, Figure 9D), suggesting a strong association between *SLC16A4* and the stromal and microenvironment components. Figure 6E further demonstrates the relationship between *SLC16A4* expression and immune cell infiltration, as assessed using the QUANTISEQ algorithm, further supporting the potential role of *SLC16A4* in immune regulation.

### 3.8. Correlation of SLC16A4 with Immune Checkpoint Gene Expression

We further analyzed the correlation between *SLC16A4* expression and the expression of immune checkpoint genes in lung cancer. As shown in Figure 10A, in the TCGA lung cancer dataset, *SLC16A4* expression was significantly positively correlated with the expression of several immune checkpoint genes. The highest correlation was observed with HAVCR2 (r = 0.326, Figure 10B), followed by SIGLEC7 (r = 0.314, Figure 10C). Additionally, BTLA and VSIR also showed relatively high correlation with *SLC16A4* (r = 0.253 and 0.228, Figure 10D,E). These results further support the potential role of *SLC16A4* in immunotherapy.

### 3.9. Correlation of SLC16A4 with Anticancer Drug Sensitivity

Using the Oncopredict package, we calculated the drug sensitivity scores for the TCGA lung cancer samples and further evaluated the correlation between *SLC16A4* expression and drug sensitivity. As shown in Figure 11A, *SLC16A4* expression was significantly associated with the sensitivity to multiple anticancer drugs, including gefitinib (r = 0.554, Figure 11B), erlotinib (r = 0.475, Figure 11C), cisplatin (r = 0.517, Figure 11D), leflunomide (r = 0.420, Figure 11E), and dactolisib (r = 0.421, Figure 11F). These results support the potential role of *SLC16A4* in influencing tumor drug sensitivity and resistance.

### 3.10. SLC16A4 Correlated with Tumor Stemness and Tumor Mutation Burden

Using lung cancer datasets from The Cancer Genome Atlas (TCGA), we investigated the association between *SLC16A4* expression and tumor stemness, tumor mutation burden (TMB), as well as microsatellite instability (MSI). *SLC16A4* expression was found to be significantly negatively correlated with tumor stemness as quantified by RNA-based stemness scores (RNAss) (r = −0.585, Figure 12A) and similarly showed a strong negative correlation with the expression of the stemness-associated marker gene SOX2 (r = −0.551, Figure 12B). To further validate this relationship, we overexpressed *SLC16A4* in lung cancer cell lines, A549 and H1299, and observed a marked reduction in SOX2 expression, as assessed by qPCR (Figure 12C,D). Additionally, correlation analyses revealed that *SLC16A4* expression was inversely associated with microsatellite instability (r = −0.291, Figure 12E) and tumor mutation burden (r = −0.200, Figure 12F).

### 3.11. SLC16A4 Correlated with Tumor-Related Biological Functions and Signaling Pathways

We further explored the potential biological significance of *SLC16A4* in lung cancer using Gene Set Enrichment Analysis (GSEA). As shown in Figure 13A,B, *SLC16A4* was associated with multiple important biological functions (|NES| > 2, adjusted *p* < 0.001), including fatty acid metabolic process (NES = 2.269), DNA replication (NES = −2.958), and recombinational repair (NES = −2.683). Additionally, enrichment analysis revealed that *SLC16A4* was associated with several cancer-related signaling pathways (Figure 13C,D), including drug metabolism cytochrome P450 (NES = 2.384), cell cycle (NES = −2.706), and mismatch repair (NES = −2.282).

### 3.12. SLC16A4 Inhibits Cell Proliferation and Promotes Cell Senescence

We investigated the effects of *SLC16A4* on lung cancer cell proliferation and senescence. CCK-8 assays showed that overexpression of *SLC16A4* significantly reduced the proliferative capacity of lung cancer cells A549 and H1299 (Figure 14A,B). EdU assays further validated that overexpression of *SLC16A4* significantly inhibited lung cancer cell proliferation (Figure 14C,D). Moreover, qPCR analysis revealed that overexpression of *SLC16A4* upregulated the expression levels of *CDKN1A* and *CDKN2A* in A549 and H1299 cells, suggesting that *SLC16A4* may suppress lung cancer cell proliferation by regulating the cell cycle (Figure 14E,F). β-galactosidase staining indicated that overexpression of *SLC16A4* promoted cell senescence in lung cancer cells (Figure 14G,H).

## 4. Discussion

In this study, we established a lung cancer diagnostic model based on serum proteomic data using four machine learning algorithms (nnet, glmnet, sgboost, and svm) and further analyzed the role of the key protein SLC16A4 in lung cancer. By analyzing the serum proteomic data from the GSE168198 dataset, we identified 23 differentially expressed proteins associated with lung cancer and constructed the diagnostic model using these four machine learning algorithms. Among these models, the XGBoost algorithm performed the best, with an AUC value of 0.985, significantly outperforming the other models (e.g., nnet with an AUC of 0.970, svm with 0.897, and glmnet with 0.861). These results indicate that serum proteomic data can serve as an effective diagnostic tool for lung cancer, and the XGBoost model, with its excellent ability to handle non-linearity and strong generalization performance, may have high accuracy in practical applications. Further evaluation on an external validation set also demonstrated that all four models exhibited strong predictive capabilities, especially svm and XGBoost, which achieved AUC values of 0.922 and 0.906, respectively, showing the potential of these machine learning models in real-world applications. In this study, we not only compared the model performance between lung cancer patients and normal controls but also introduced a disease control group to more comprehensively assess the diagnostic capability of the model. The inclusion of the disease control group provided us with an important reference framework, allowing us to more clearly identify the sensitivity and specificity of the model within specific disease contexts. As highlighted in the literature, evaluating model performance with disease controls can reveal the behavior of potential biomarkers across different pathological states [22].

After comparing with other previously reported tumor markers, our study further validates the superiority of the machine learning model in the early diagnosis of lung cancer. Increasingly, machine learning algorithms are being used to train models for predicting and assisting in disease diagnosis, thus improving diagnostic accuracy [23]. Previous studies based on blood metabolites have used various machine learning algorithms to develop early lung cancer prediction models, with the XGBoost model demonstrating excellent predictive capability (AUC = 0.81) [24]. Another study utilized DNA copy number variations in plasma to construct a lung cancer predictor using XGBoost, which is also a non-invasive analytical method [25]. A recent study used plasma-free immune-related miRNAs from a large sample cohort to build an early cancer diagnostic model, which showed outstanding predictive performance in various cancers, especially in lung and gastric cancers, with AUC values reaching 0.990 [26].

Based on the importance of proteins in the model, we further focused on SLC16A4 for additional analysis. Existing studies have indicated that members of the SLC16 family are involved in a wide range of metabolic pathways, including energy metabolism in the brain, skeletal muscle, heart, and tumor cells, as well as intestinal metabolism and drug transport [27]. Furthermore, SLC16 family proteins play a crucial role in maintaining the intracellular concentrations of various important endogenous molecules in health and disease, establishing them as a novel therapeutic target in cancer [28,29]. Our study found that the expression of *SLC16A4* at both the mRNA and protein levels was significantly lower in lung cancer tissues compared to normal tissues. Moreover, the expression of *SLC16A4* was closely associated with the clinical pathological features of lung cancer patients, including gender, tumor invasion depth, lymph node metastasis, tumor stage, and smoking history. These findings suggest that the expression of *SLC16A4* may be closely related to the initiation and progression of lung cancer and could potentially serve as a biomarker for evaluating lung cancer progression and prognosis. Correlation analysis further showed that the expression of *SLC16A4* was associated with the infiltration of various immune cells, suggesting that *SLC16A4* might participate in the occurrence and development of lung cancer by regulating immune responses in the tumor microenvironment. Furthermore, *SLC16A4* expression was positively correlated with the expression of multiple immune checkpoint genes, further supporting its potential role in immune escape.

In vitro experiments revealed that overexpression of *SLC16A4* significantly inhibited the proliferation of A549 and H1299 lung cancer cells and promoted cellular senescence. Cellular senescence is an irreversible cell-cycle arrest state induced by various stresses and acts as a cellular defense mechanism to prevent the accumulation of damage and inhibit potential tumorigenesis. Although senescence is considered a factor promoting age-related diseases, it also plays a key role in cancer defense by preventing tumorigenesis through the induction of cellular senescence, making it a potential therapeutic target in cancer treatment [30,31,32]. Further research into the mechanisms of *SLC16A4* in cell proliferation, senescence, and immune escape will help elucidate its potential as an anti-tumor target.

Despite the strong performance of the lung cancer diagnostic model developed in this study based on current data, there are still some limitations. First, this study mainly relied on data from public databases and did not fully validate the model’s accuracy in a broader population. Second, the expression and clinical significance of *SLC16A4* require further validation with more clinical data. Moreover, the specific mechanisms of *SLC16A4* remain unclear, and further studies on its role in cell proliferation, senescence, and immune escape will help advance its application as a therapeutic target.

## 5. Conclusions

In conclusion, this study constructed a lung cancer diagnostic machine learning model using serum proteomic data from lung cancer patients, demonstrating strong diagnostic value. The key protein SLC16A4 in the model not only serves as a potential biomarker for lung cancer but may also play an important role in lung cancer immunotherapy and targeted therapy. Through in-depth research into the biological functions of SLC16A4, it is expected to provide new insights for early lung cancer diagnosis and personalized treatment in the future.

## 6. Limitations

This study has several limitations. Firstly, there is a significant disparity in the number of proteins detected between the two platforms used (GPL29810 detected 89 proteins, while GPL29809 detected 16,048 proteins). Although we standardized the data to ensure model applicability across platforms, this discrepancy could impact model performance and protein feature selection. Secondly, the external validation set has a limited sample size of only 16 samples. While these samples provide an independent validation pathway, the small sample size might not sufficiently evaluate the model’s generalizability to a broader population. Future studies should incorporate larger and more diverse external datasets to further validate and optimize the model. Lastly, our data source is solely from the GEO database (GSE168198). Despite rigorous selection and preprocessing, inherent limitations and biases in the database might affect the study results. To enhance model validation, future research should include independent public datasets and multi-center clinical data.

## Figures and Tables

**Figure 1 biomolecules-15-01081-f001:**
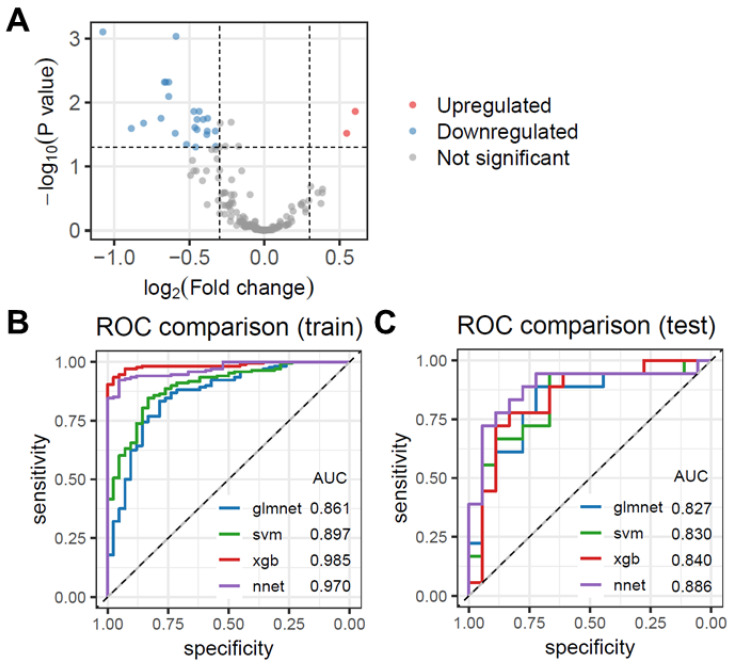
Diagnostic models were constructed based on four machine learning algorithms using the training set. (**A**) Differentially expressed proteins in the plasma of lung cancer patients from the GSE168198 dataset were analyzed. (**B**) ROC curves display the performance of diagnostic models constructed using the four machine learning algorithms. (**C**) ROC curves show the performance of the four models in the internal testing set.

**Figure 2 biomolecules-15-01081-f002:**
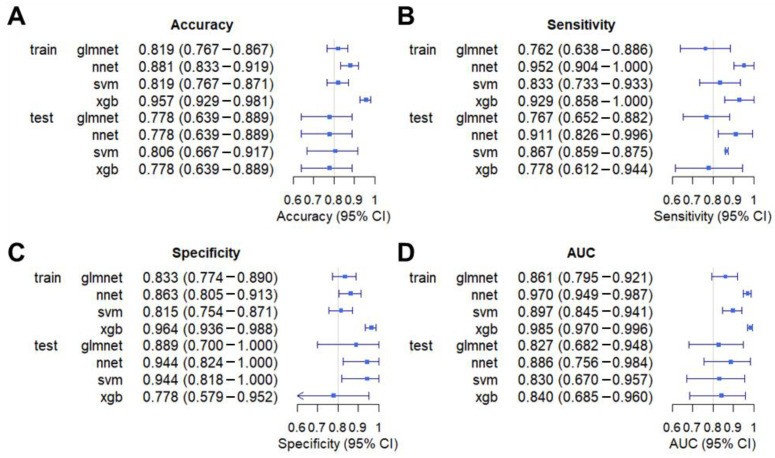
Forest plot showing the evaluation results of the models. The forest plots, respectively, display the Accuracy (**A**), Sensitivity (**B**), Specificity (**C**), and AUC (**D**) of four models in the training and test datasets.

**Figure 3 biomolecules-15-01081-f003:**
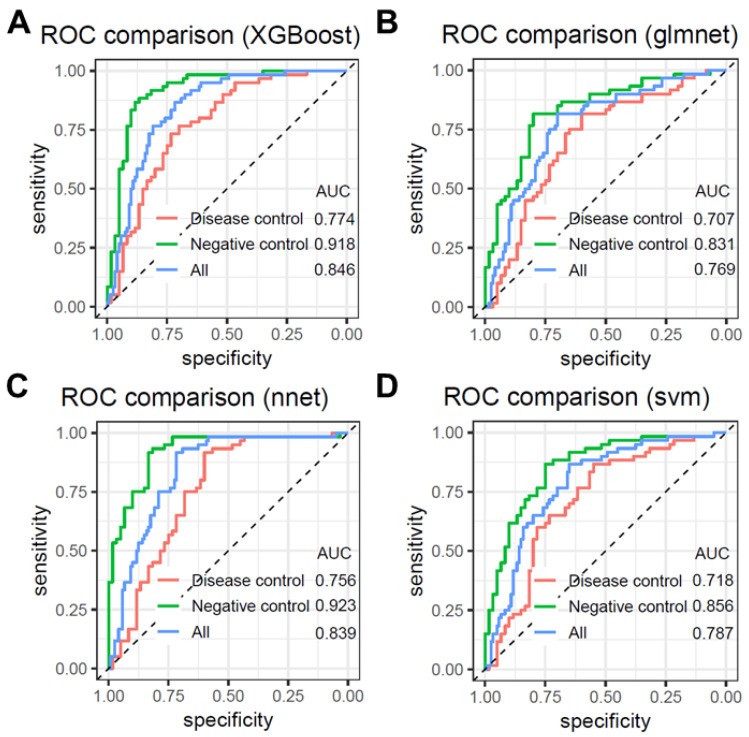
The predictive performance of the four models was analyzed in different control datasets. (**A**) XGBoost, (**B**) glmnet, (**C**) nnet, (**D**) svm.

**Figure 4 biomolecules-15-01081-f004:**
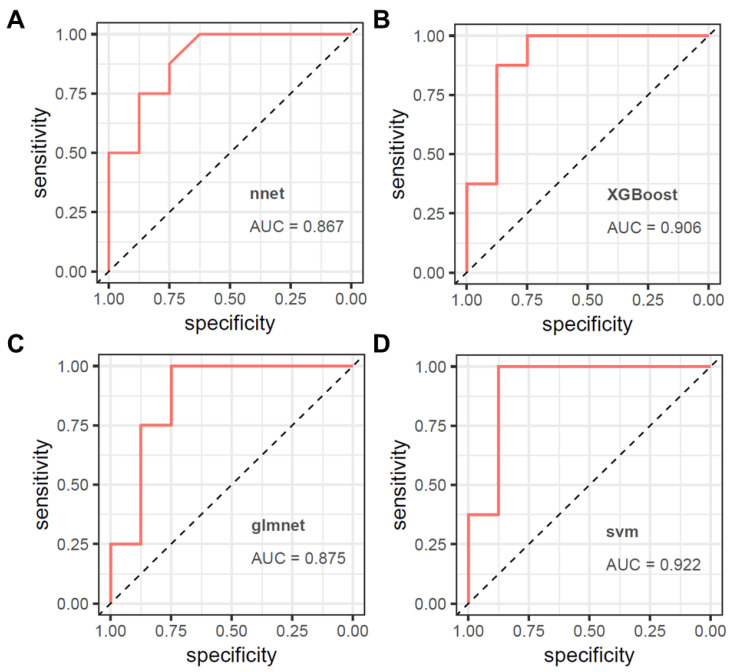
The predictive performance of the four models was analyzed in the external testing set. (**A**) XGBoost, (**B**) glmnet, (**C**) nnet, (**D**) svm.

**Figure 5 biomolecules-15-01081-f005:**
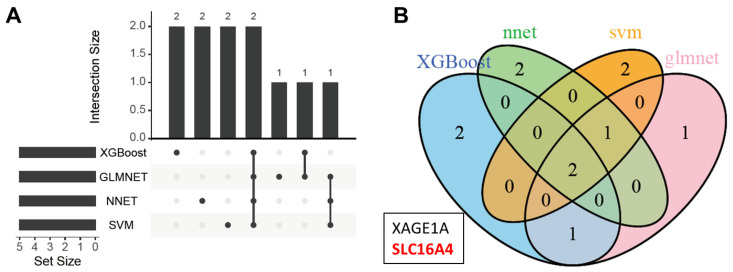
The intersection of the top 5 important proteins identified across the four models. (**A**) Upset plot. (**B**) Venn diagram. The number in venn diagram represents the intersection of the top 5 important genes identified by different algorithms.

**Figure 6 biomolecules-15-01081-f006:**
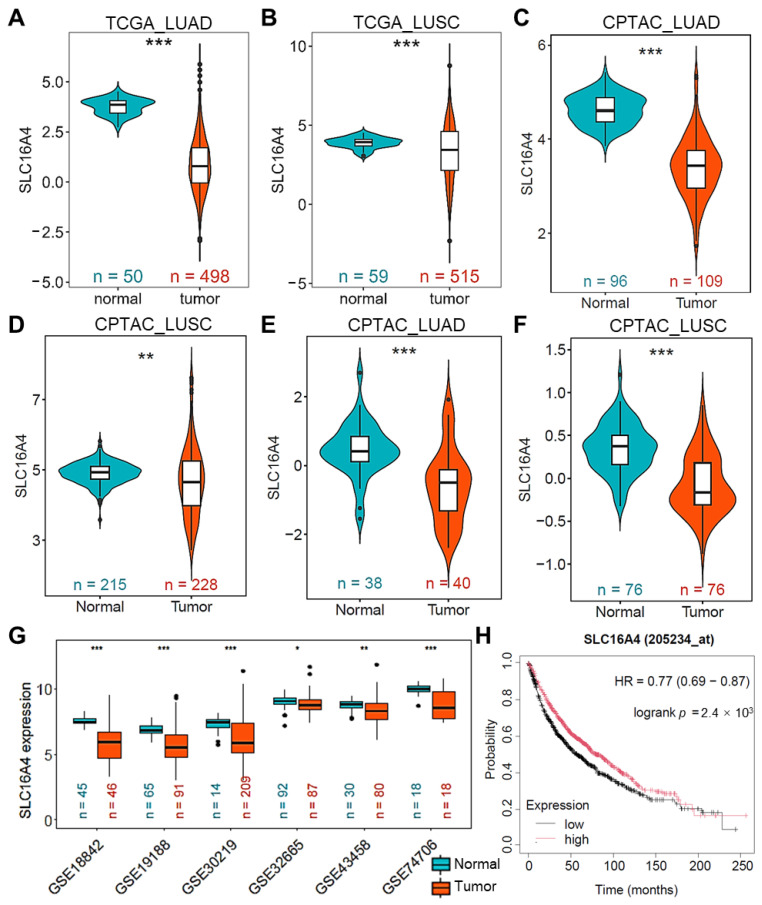
SLC16A4 is significantly downregulated in lung cancer. The mRNA expression differences of *SLC16A4* between lung cancer and normal tissues in the TCGA LUAD (**A**) and LUSC (**B**) datasets; in the CPTAC LUAD (**C**) and LUSC (**D**) datasets; and in protein expression in the CPTAC LUAD (**E**) and LUSC (**F**) datasets. (**G**) mRNA expression differences of *SLC16A4* in lung cancer and normal tissues across multiple lung cancer datasets from the GEO database. (**H**) The effect of *SLC16A4* on overall survival in lung cancer patients was analyzed using the KMPlotter online tool. Compared with normal group, * *p* < 0.05, ** *p* < 0.01, *** *p* < 0.001. TCGA, The Cancer Genome Atlas. CPTAC, Clinical Proteomic Tumor Analysis Consortium. LUAD, Lung Adenocarcinoma. LUSC, Lung Squamous Cell Carcinoma. GEO, Gene Expression Omnibus.

**Figure 7 biomolecules-15-01081-f007:**
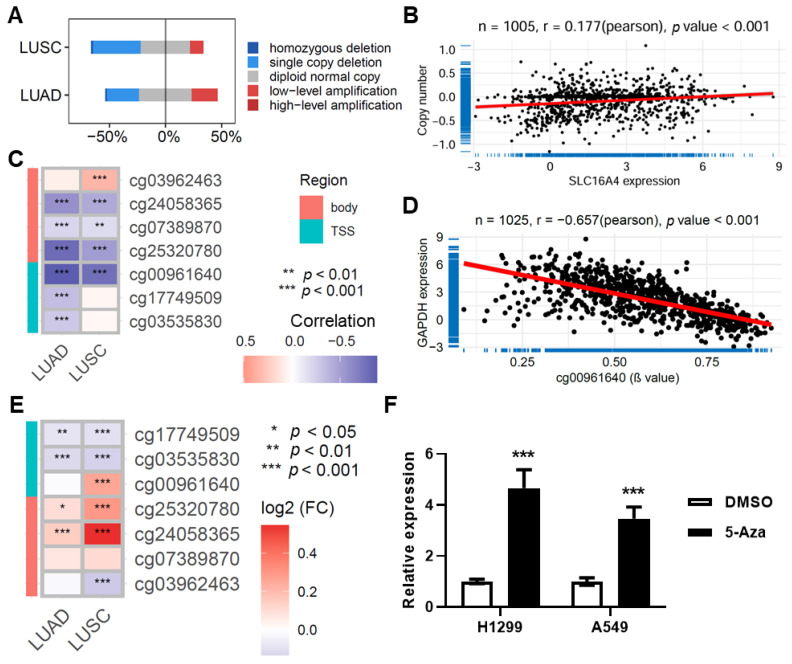
Regulation of *SLC16A4* by copy number variation and DNA methylation. (**A**) Distribution of DNA copy number variations in the lung cancer dataset based on TCGA. (**B**) Scatter plot showing the correlation between *SLC16A4* expression and DNA copy number in the TCGA lung cancer dataset. (**C**) Heatmap displaying the correlation between *SLC16A4* expression and methylation levels in the lung cancer dataset. (**D**) Scatter plot showing the correlation between *SLC16A4* expression and the methylation level of the cg00961640 site in the TCGA lung cancer dataset. (**E**) The heatmap shows the differential methylation levels at various sites in tumor tissues from the lung cancer dataset in TCGA. (**F**) qPCR analysis of *SLC16A4* expression changes in A549 and H1299 cells following treatment with the methylation inhibitor 5-aza. Compared with DMSO, *** *p* < 0.001. TCGA, The Cancer Genome Atlas. LUAD, Lung Adenocarcinoma. LUSC, Lung Squamous Cell Carcinoma.

**Figure 8 biomolecules-15-01081-f008:**
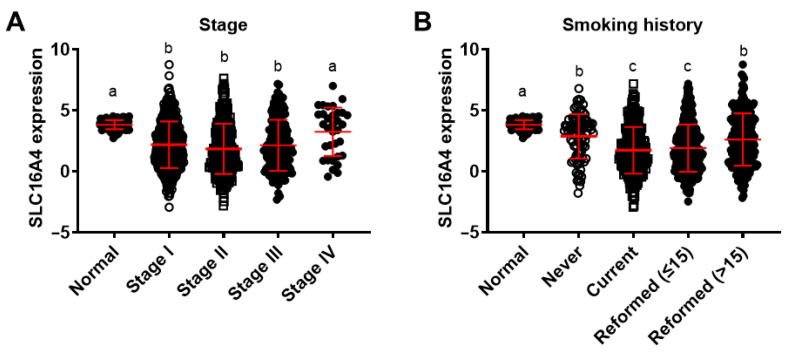
Differences in *SLC16A4* expression in tumors from patients with different stages and smoking histories. (**A**) Tumor staging; (**B**) Smoking history. The two groups do not contain the same letters indicating a significant difference, *p* < 0.05.

**Figure 9 biomolecules-15-01081-f009:**
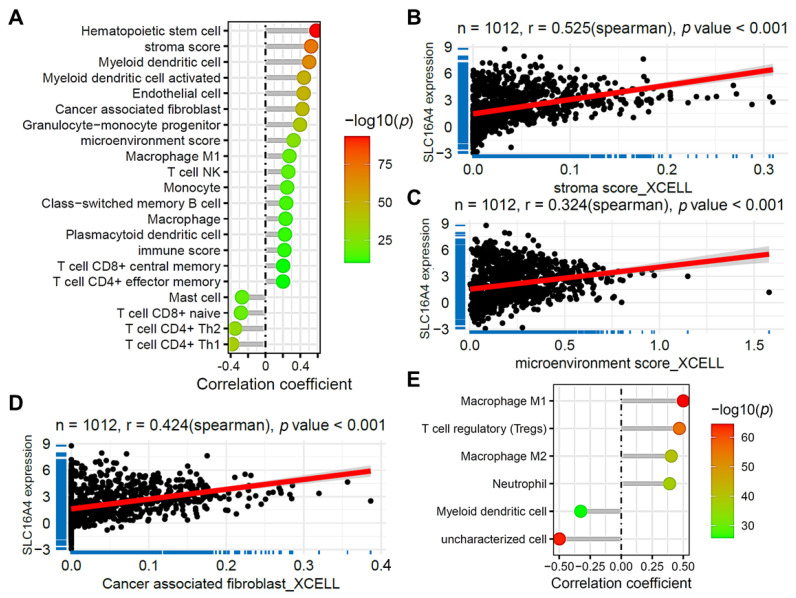
Association of *SLC16A4* with immune cell infiltration. (**A**) Lollipop plot showing the correlation between *SLC16A4* expression and the infiltration of various immune and stromal cells calculated using the XCELL algorithm in the TCGA lung cancer dataset. (**B**) Scatter plot showing the correlation between *SLC16A4* expression and stromal score (**B**), microenvironment score (**C**), and cancer-associated fibroblast infiltration (**D**) based on the XCELL algorithm. (**E**) Lollipop plot demonstrating the correlation between *SLC16A4* expression and immune cell infiltration, calculated using the QUANTISEQ algorithm in the TCGA lung cancer dataset. TCGA, The Cancer Genome Atlas.

**Figure 10 biomolecules-15-01081-f010:**
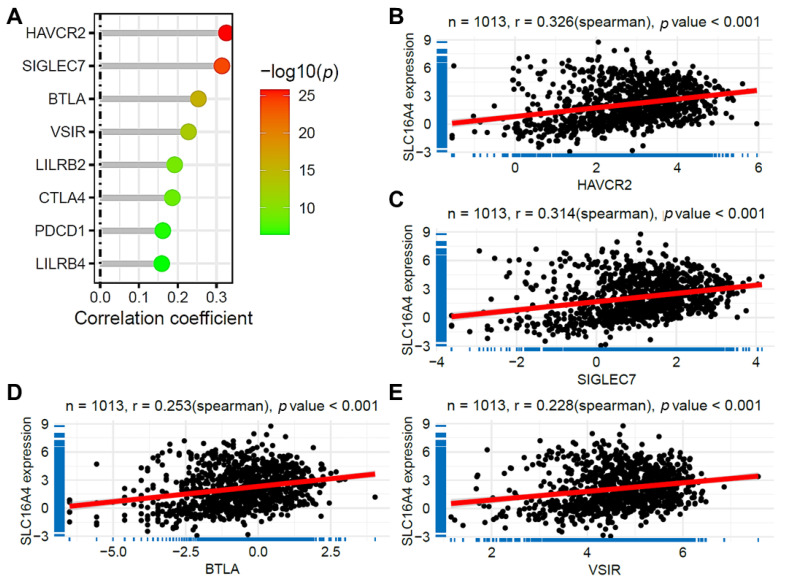
Association of *SLC16A4* with immune checkpoint genes. (**A**) Lollipop plot showing the correlation between *SLC16A4* expression and the expression of immune checkpoint genes in the TCGA lung cancer dataset. (**B**) Scatter plots showing the correlation between *SLC16A4* expression and HAVCR2 (**B**), SIGLEC7 (**C**), BTLA (**D**), and VSIR (**E**). TCGA, The Cancer Genome Atlas.

**Figure 11 biomolecules-15-01081-f011:**
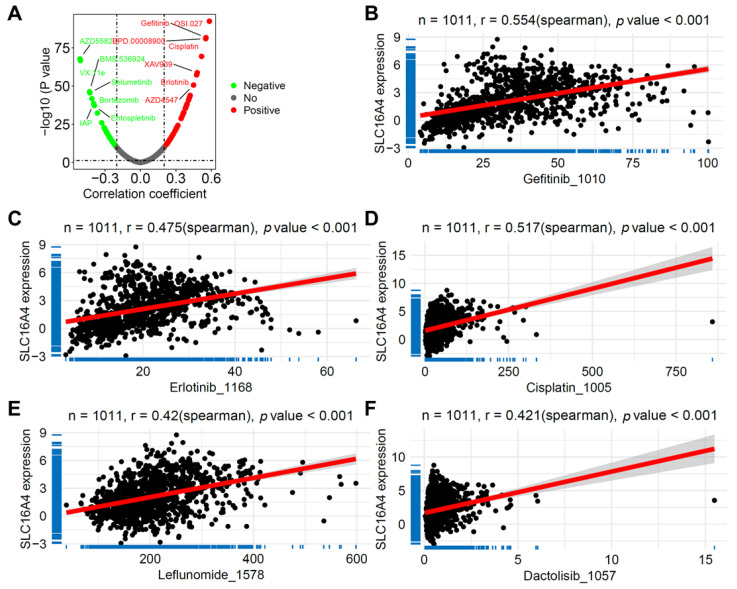
Association of *SLC16A4* with anticancer drug sensitivity. (**A**) Lollipop plot showing the correlation between *SLC16A4* expression and anticancer drug sensitivity in the TCGA lung cancer dataset. (**B**) Scatter plots showing the correlation between *SLC16A4* expression and the sensitivity to gefitinib (**B**), erlotinib (**C**), cisplatin (**D**), leflunomide (**E**), and dactolisib (**F**). TCGA, The Cancer Genome Atlas.

**Figure 12 biomolecules-15-01081-f012:**
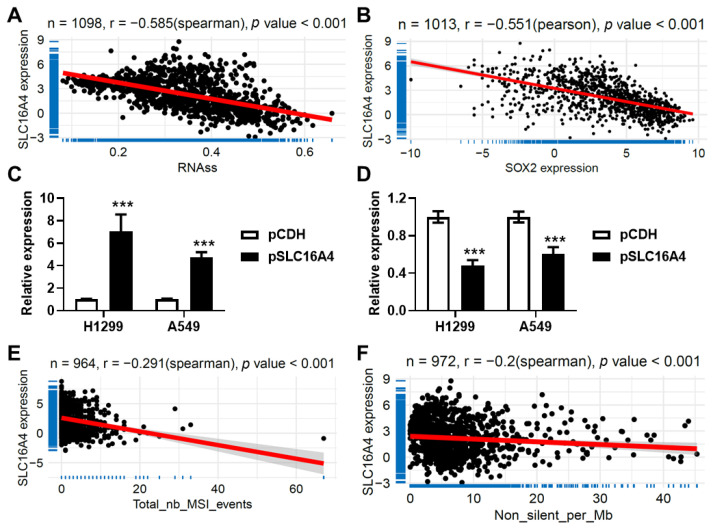
*SLC16A4* is associated with tumor stemness and tumor mutation burden. Scatter plots showing the correlation between *SLC16A4* expression and RNA-based stemness scores (**A**) and *SOX2* expression (**B**), respectively. qPCR analysis of *SLC16A4* (**C**) and *SOX2* (**D**) mRNA levels in A549 and H1299 cells transfected with *SLC16A4* overexpression plasmids. Compared with pCDH, *** *p* < 0.001. Scatter plots illustrating the correlation between *SLC16A4* expression and microsatellite instability scores (**E**) and tumor mutation burden (**F**).

**Figure 13 biomolecules-15-01081-f013:**
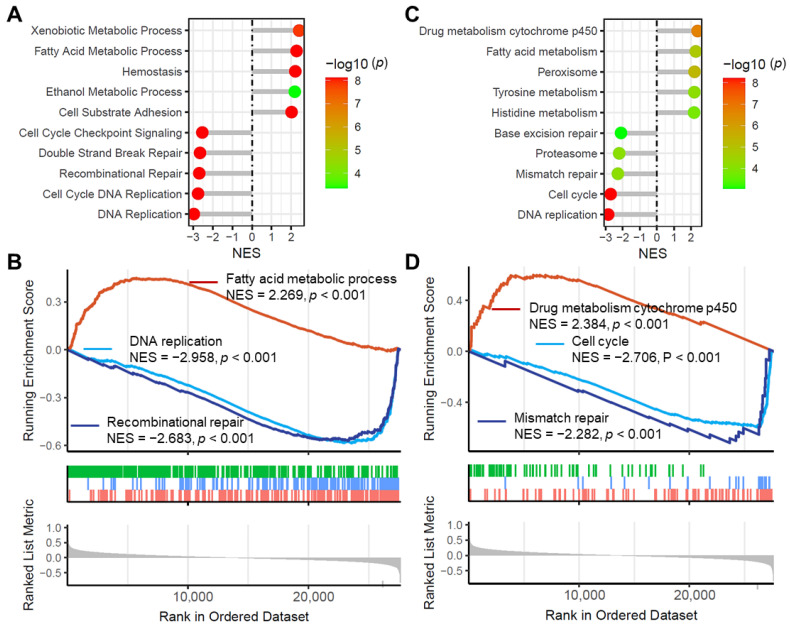
Association of *SLC16A4* with tumor-related biological functions and signaling pathways. Lollipop plots and GSEA plots showing the GSEA analysis of *SLC16A4* based on the TCGA lung cancer dataset: (**A**,**B**) biological functions and (**C**,**D**) KEGG signaling pathways. GSEA, Gene Set Enrichment Analysis. TCGA, The Cancer Genome Atlas. KEGG, Kyoto Encyclopedia of Genes and Genomes.

**Figure 14 biomolecules-15-01081-f014:**
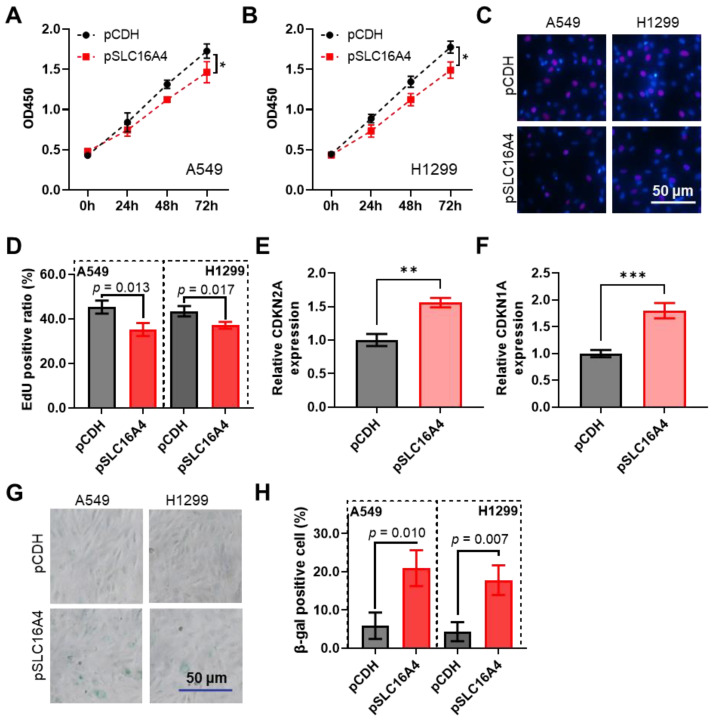
*SLC16A4* inhibits lung cancer cell proliferation and promotes cell senescence. CCK-8 assays show the change in proliferative capacity of lung cancer cells A549 (**A**) and H1299 (**B**) following *SLC16A4* overexpression. EdU assays show representative images (**C**) and statistical results (**D**) of proliferation in A549 and H1299 cells after *SLC16A4* overexpression. qPCR analysis reveals changes in *CDKN1A* (**E**) and *CDKN2A* (**F**) expression in lung cancer cells after *SLC16A4* overexpression. Compared with pCDH. β-galactosidase staining shows representative images (**G**) and statistical results (**H**) of cell senescence following *SLC16A4* overexpression in lung cancer cells. * *p* < 0.05, ** *p* < 0.01, *** *p* < 0.001.

**Table 1 biomolecules-15-01081-t001:** Distribution of samples in the training set, testing set, and validation set.

Characteristics		Training (*n* = 144)	Testing (*n* = 36)	Validating (*n* = 16)	All (*n* = 196)	*p*
Age	>60	56 (42.75%)	14 (41.18%)	4 (25.00%)	74 (40.88%)	0.602
	≤60	75 (57.25%)	20 (58.82%)	12 (75.00%)	107 (59.12%)	
gender	female	49 (36.84%)	18 (50.00%)	10 (62.50%)	77 (41.62%)	0.161
	male	84 (63.16%)	18 (50.00%)	6 (37.50%)	108 (58.38%)	
smoke	no	24 (68.57%)	11 (84.62%)	8 (100.00%)	43 (76.79%)	0.242
	yes	11 (31.43%)	2 (15.38%)	0	13 (23.21%)	
surgery	no	2 (5.71%)	1 (7.69%)	0	3 (5.36%)	0.896
	yes	33 (94.29%)	12 (92.31%)	8 (100.00%)	53 (94.64%)	
Disease type	Negative Control	42 (29.17%)	18 (50.00%)	8 (50.00%)	68 (34.69%)	NA
	Disease control	60 (41.67%)	0	0	60 (30.61%)	
	Cancer	42 (29.17%)	18 (50.00%)	8 (50.00%)	68 (34.69%)	

Note: Due to missing demographic and clinical data for some patients as provided by the GEO dataset authors, the sum of individuals in specific categories may not equal the total sample size. NA, Not Applicable.

**Table 2 biomolecules-15-01081-t002:** Correlation of *SLC16A4* expression with clinical characteristics in the TCGA lung cancer dataset.

Characteristics		N	Expression	*p*
Gender	female	405	2.580 ± 1.879	<0.001
	male	608	1.812 ± 2.026	
Age	>60	720	2.121 ± 2.014	0.768
	≤60	265	2.079 ± 1.986	
M	M0	753	2.049 ± 2.023	0.002
	M1	32	3.241 ± 2.021	
N	N0	648	2.103 ± 1.973	0.041
	N1	226	1.871 ± 2.013	
	N2	114	2.530 ± 2.119	
	N3	7	2.073 ± 1.375	
T	T1	282	2.505 ± 1.907	0.002
	T2	569	1.968 ± 1.976	
	T3	118	1.911 ± 2.241	
	T4	41	1.995 ± 1.992	
Stage	Stage I	518	2.186 ± 1.914	0.001
	Stage II	283	1.853 ± 2.053	
	Stage III	167	2.141 ± 2.089	
	Stage IV	33	3.246 ± 1.989	
Smoke.history	Current	253	1.736 ± 1.910	<0.001
	Never	94	2.877 ± 1.819	
	Reformed (>15)	215	2.615 ± 2.149	
	Reformed (≤15)	416	1.914 ± 1.938	

## Data Availability

Publicly available datasets were analyzed in this study. The TCGA BLCA data can be accessed from The Cancer Genome Atlas (TCGA) at https://portal.gdc.cancer.gov/, accessed on 15 March 2025. The GEO dataset can be found in the Gene Expression Omnibus (GEO) database at https://www.ncbi.nlm.nih.gov/geo, accessed on 15 March 2025.

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
