# Peer review of "Development of a Serum Proteomic-Based Diagnostic Model for Lung Cancer Using Machine Learning Algorithms and Unveiling the Role of SLC16A4 in Tumor Progression and Immune Response"

_biomolecules, 2025, doi:10.3390/biom15081081_

Round 1
Reviewer 1 Report
Comments and Suggestions for Authors
It needs to be clarified what is the practical implication at the clinical practice level
SLC16A4 expression seems to be protective. The differences between training and testing set should be better explained. It needs to be clarified how association of SLC16A4 expression with other variables was assessed
Please, add the limitations of the study e.g. numerical dissimilarity of the two subgroups. the statistical analysis section should clarify which were the comparison groups for t-test and which for ANOVA. The characteristics of the control group must be added, e.g., comorbidities or any previous malignancies
A comparative discussion should be done comparing it with other biomarkers
in this regard
I suggest including the following reference: Thorac Cancer. 2020 Nov;11(11):3060-3070.
Author Response
Comments 1: It needs to be clarified what is the practical implication at the clinical practice level.
Response 1: Our study developed a lung cancer diagnostic model based on serum proteomics data, utilizing four machine learning algorithms (nnet, glmnet, svm, and XGBoost). This model aims to provide a non-invasive tool for the early diagnosis of lung cancer. Early diagnosis is crucial for the prognosis of lung cancer patients and can significantly improve treatment outcomes and survival rates. Our model has been validated on multiple datasets, including normal controls, disease controls, and lung cancer data, demonstrating good diagnostic performance. Furthermore, additional validation on an independent external dataset indicates that the model has broad applicability and reliability. We believe this diagnostic model has the potential to be used in clinical practice for screening and early detection of lung cancer, thereby improving patient management and treatment decisions.
Comments 2: SLC16A4 expression seems to be protective.
Response 2: Our study found that the expression of SLC16A4 is significantly downregulated in lung cancer serum samples, and its overexpression can significantly inhibit lung cancer cell proliferation and induce cellular senescence, suggesting that SLC16A4 may have a protective role in lung cancer development. This finding indicates that SLC16A4 is not only a potential diagnostic biomarker but may also serve as a therapeutic target for further research.
Comments 3: The differences between training and testing set should be better explained.
Response 3: Thank you for your valuable feedback on our manuscript. We are pleased to address your question and provide further clarification regarding the differences between the training set and the testing set:
- Training Set and Internal Testing Set: Both the training set and the internal testing set in this study were derived from the GSE168198 dataset, utilizing the BCBIO HUMAN CUSTOM PROTEOME MICROARRAY platform (platform annotation file GPL29810). This platform can detect 89 plasma proteins. The GSE168198 dataset contains plasma sample data from 180 samples analyzed with this chip. These data were used to train and internally validate our lung cancer diagnostic model. We employed cross-validation methods to ensure the robustness and generalizability of the model on both the training set and the internal testing set.
- External Validation Set: To further validate the model's performance, we utilized an external validation set. The external validation set used the CDI LABS HUMAN PROTEOME MICROARRAY v4.0 platform (platform annotation file GPL29809), which can detect 16,048 proteins. Within the GSE168198 dataset, there are 16 plasma samples analyzed with this chip. Due to differences in the number and types of proteins detected by the different chip platforms, we standardized the data to ensure the model's applicability and accuracy across different platforms.
We understand that the differences between the training and testing sets may raise concerns, and therefore, we implemented strict standardization and cross-validation methods in the data processing and model validation phases to ensure the reliability and consistency of the results. These differences do not significantly affect the overall performance of the model; rather, they enhance the model’s generalization capability across different datasets and platforms, thereby increasing its potential for clinical application. We have elaborated on these issues in the revised manuscript section 2.1 dataset to further enhance the transparency and scientific rigor of the study. Thank you for your attention and suggestions regarding our work.
Comments 4: It needs to be clarified how the association of SLC16A4 expression with other variables was assessed.
Response 4: Thank you for your in-depth review and the questions posed regarding our paper. I would like to clarify that the primary aim of this study is to construct a machine learning-based lung cancer diagnostic model, with SLC16A4 being regarded as a key molecule selected for our model.
Given that our focus is on model construction and performance evaluation rather than a separate analysis of SLC16A4's association with other clinical variables, we did not elaborate on this aspect in the study. We believe that the selection of SLC16A4 was based on its importance within the model, rather than a direct correlation analysis with other variables.
Thank you again for your attention and understanding of our research. If you have any further suggestions or comments, we are more than willing to consider them.
Comments 5: The limitations of the study e.g. numerical dissimilarity of the two subgroups.
Response 5: Thank you for your valuable suggestions regarding our manuscript. We understand the importance of discussing limitations in research and are willing to further clarify and supplement relevant content to enhance the transparency and comprehensiveness of our study.
In our research, there are indeed several limitations that need to be pointed out:
- Dissimilarity in subgroup sample sizes: There is a significant difference in the number of proteins detected between the two platforms (GPL29810 and GPL29809) we used. Specifically, the GPL29810 platform detected 89 proteins, while the GPL29809 platform detected 16,048 proteins. Although we standardized the data to ensure the model's applicability across different platforms, this difference may affect the model's performance and the selection of protein features.
- Limited sample size: Particularly in the external validation cohort, the sample size is small (only 16 samples). While these samples provide an independent validation pathway for the model, the limited sample size may be insufficient to comprehensively assess the model's generalizability to a broader population. Future studies will require more external datasets and larger sample sizes to further validate and optimize our model.
- Limitations of the dataset: Our study data are all sourced from the GEO database (GSE168198). Although these data have undergone rigorous screening and processing, the limitations of the database itself and biases in the data collection process may impact the study results. To further validate the model's effectiveness, we recommend incorporating other independent public datasets and multi-center clinical data in future research.
We will discuss these limitations in detail in the revised manuscript and propose directions for future research to further enhance the robustness and broad applicability of the model. Thank you for your attention to our work and your constructive suggestions.
Comments 6: The statistical analysis section should clarify which were the comparison groups for t-test and which for ANOVA.
Response 6: In the original manuscript, we failed to clearly specify which comparison groups utilized the t-test and which utilized ANOVA. We have made the necessary additions based on your suggestions to ensure the transparency and accuracy of the statistical analysis. Specifically, comparisons between two groups primarily include gene expression levels in tumor and normal samples, differential analysis of gene expression between the 5-aza treatment and NC groups in the DNA methylation validation experiments, differential gene expression between the SLC16A4 overexpression group and the empty vector transfection group, as well as gene expression differences in clinical pathological features containing only two categories; multi-group comparisons mainly involve gene expression differences related to clinical pathological features with more than two categories.
Comments 7: The characteristics of the control group must be added, e.g., comorbidities or any previous malignancies.
Response 7: Thank you for your review of our paper and your valuable suggestions. In response to your recommendation to add characteristics of the control group (such as comorbidities or history of previous malignancies), we have conducted further analysis and explanation.
Comorbidity information: We have detailed the specific statistical data on different disease types in the control group in Table 1, including the distribution of various comorbidities. This information helps to comprehensively present the characteristics of the control group, ensuring the reliability of the research results.
History of previous malignancies: Unfortunately, the GSE168198 dataset does not provide information regarding the history of previous malignancies. Therefore, we are unable to include this characteristic in our analysis. However, we believe that even in the absence of this information, our multi-omics analysis combined with various machine learning algorithms can still effectively establish an efficient lung cancer diagnostic model and reveal the important role of SLC16A4 in lung cancer.
We hope these additions and explanations will meet your requirements, and we look forward to further feedback.
Comments 8: A comparative discussion should be done comparing it with other biomarkers in this regard.
Response 8: Thank you for your review of our paper and your valuable suggestions. Based on your feedback, we have included a comparative discussion with other reported tumor biomarkers in the manuscript to more comprehensively showcase the relative advantages and significance of our research.
Specifically, we have added a comparison of the diagnostic efficacy of nine reported tumor biomarkers, with detailed information provided in Supplementary Table S3. These biomarkers include DKK1, MYC, TP53, MUC1, SOX2, EZR, HDAC1, YAP1, and HER2. In our study, the AUC values for these biomarkers did not exceed 0.70, while our machine learning model achieved AUC values exceeding 0.80, significantly outperforming single biomarkers. These results further validate the efficiency and accuracy of our model.
We believe that these additions and explanations can adequately demonstrate the advantages of our research, and we hope these improvements will meet your requirements.
Comments 9: I suggest including the following reference: Thorac Cancer. 2020 Nov;11(11):3060-3070.
Response 9: Thank you for your valuable suggestions and for reviewing our paper. Based on your feedback, we have added the reference you recommended to further support our discussion.
Once again, thank you for your attention and guidance regarding our work.
Reviewer 2 Report
Comments and Suggestions for Authors
The authors present an analysis of a serum proteomic-based diagnostic model, using machine learning with a focus on SLC16A4. Whilst the manuscript is generally well-written, there are some gaps in the methods and results.
[1] Lines 102-103, the authors say they will use AUROC, accuracy, sensitivity, specificity, precision and F1 score to assess the four models. But they do not actually do this, or I cannot see this assessment in the results? Could the authors signpost to how these measures are used, and where in the manuscript?
[2] Line 122, the authors state that they found 4 proteins common across the models, but later discuss 2 proteins. Could the authors please explain the difference?
[3] Where are the methods for the proteomic analysis of the 10 people in the validation set? If the authors did a proteomics analysis and then applied the models to this external cohort, where are these data from, exactly? It would be helpful to include explanations in Methods.
[4] In results, Table 1, the numbers appear to be problematic. The training set contains 144 people, and has 56 over 60 years old and 75 under 60 years old. Could the authors clarify what happened to the other people?
[5] There are many other issues with Table 1, I will not list them all, but none of the numbers add up to the correct n. Could the authors ensure that Table 1 is correct?
[6] Table 1 says that there are 16 people in the validation set, but line 258 says there are 10 people in the validation set? Could the authors ensure that the data are consistent?
[7] Line 260, contrary to what is said elsewhere, SVM appears to be the best performing algorithm in the validation set? Could the authors make it more clear which they think is the best model and exactly what criteria they used?
[8] The authors could usefully consider an UpSet plot instead of a Venn diagram.
[9] It was not clear to me why all of the results focus on SLC164 and not XAGE1A. Both proteins were common to all 4 models, surely both should be analysed here? Or could the authors make it more clear why one was selected and not other?
[10] The referencing in Discussion could be improved. For example, the authors include in Results the three-class model performance between cancer, healthy controls, and also disease controls. This is an important distinction, as highlighted by this work (PMID: 38076065), so the authors should consider usefully including this reference (or an equivalent citation that explains the importance of disease controls) and focus on the three-class model performance.
Similarly, these sorts of analysis have been done before, and SLC16 family has been often linked to cancer. Could the authors expand the Discussion with a comprehensive analysis of how their findings are consistent or contrast with other research on SLC16? At the moment there is only one reference to the SLC16 family, the authors should consider adding PMID: 32144120 or PMID: 36706846 amongst others.
[11] The data availability statement could usefully specify the exact datasets used, not just link to the TCGA data or the GEO data. The data availability statement should also address the data from the validation cohort and the underlying data for the assays. Is this available online or is this available on request?
[12] As a final comment, which applies throughout, the models will have produced confusion matrices for diagnostic models. Confidence intervals can then be calculated for accuracy, sensitivity, specificity and so on, for example using the epiR library in R. This is important because there is a confidence interval associated with any binary diagnostic test. A coin will be heads 50% of the time, as the "true" value, but if you toss a coin 100 times it is unlikely that exactly 50 heads will be produced, but 0.50 will probably be in the 95% confidence interval. Please can the authors add 95% confidence intervals to all model metrics.
Author Response
Comments [1] Lines 102-103, the authors say they will use AUROC, accuracy, sensitivity, specificity, precision and F1 score to assess the four models. But they do not actually do this, or I cannot see this assessment in the results? Could the authors signpost to how these measures are used, and where in the manuscript?
Response 1: Thank you for pointing out the inconsistency between the statement regarding assessment metrics in lines 102-103 and the results description. We did conduct multiple evaluations of the models in the manuscript; however, we failed to comprehensively present all the metrics in the results section. We have revised the manuscript to include the results for each assessment metric. We hope these modifications will provide a clearer presentation of all assessment metrics and enhance the completeness of the manuscript.
Comments [2] Line 122, the authors state that they found 4 proteins common across the models, but later discuss 2 proteins. Could the authors please explain the difference?
Response 2: Thank you for pointing out the discrepancy regarding the number of proteins in line 122. We confirm that there was an error (a typographical mistake) in that description. In fact, we discovered and discussed 2 proteins, not 4. We have revised the manuscript to ensure accurate representation.
Comments [3] Where are the methods for the proteomic analysis of the 10 people in the validation set? If the authors did a proteomics analysis and then applied the models to this external cohort, where are these data from, exactly? It would be helpful to include explanations in Methods.
Response 3: Thank you for raising the question about the methods for proteomic analysis and the data sources for the validation set. We will provide a more detailed explanation and will include relevant content in the Methods section.
Detailed Explanation:
Regarding the differences between the training set and the testing set, we provide a more detailed explanation here:
- Training Set and Internal Testing Set: The training and internal testing sets in this study were both derived from the GSE168198 dataset, utilizing the BCBIO HUMAN CUSTOM PROTEOME MICROARRAY chip (platform annotation file GPL29810). This chip can detect 89 plasma proteins. The GSE168198 dataset includes plasma sample data from 180 individuals analyzed with this chip. These data were used to train and internally validate our lung cancer diagnostic model. We employed cross-validation methods to ensure the robustness and generalizability of the model on the training and internal testing sets.
- External Validation Set: To further validate the model's performance, we utilized an external validation set. This external validation set used the CDI LABS HUMAN PROTEOME MICROARRAY v4.0 chip (platform annotation file GPL29809), which can detect 16,048 proteins. Within the GSE168198 dataset, there are 16 plasma samples analyzed using this chip. Due to differences in the number and types of proteins detected by different chip platforms, we standardized the data to ensure the model's applicability and accuracy across different platforms.
We will include the above information in the Methods section so that readers can have a more comprehensive understanding of our data processing and model validation methods. Thank you once again for your careful review and valuable suggestions.
Comments [4] In results, Table 1, the numbers appear to be problematic. The training set contains 144 people, and has 56 over 60 years old and 75 under 60 years old. Could the authors clarify what happened to the other people?
Response 4: Thank you for your careful review of Table 1 in our manuscript and for your question.
Regarding the demographic data in the training set, we would like to clarify:
According to the sample information provided by the authors of the GEO dataset, some patients lack certain demographic and clinical data. This resulted in only a subset of samples containing complete demographic information when we conducted data statistics. Therefore, the sum of patients over 60 and under 60 in the training set (56 and 75, totaling 131) does not cover all 144 samples.
We have added a note below Table 1 in the manuscript to ensure readers understand the reason for the incomplete data and how it was handled. (Note: Due to missing demographic and clinical data for some patients as provided by the GEO dataset authors, the sum of individuals in specific categories may not equal the total sample size.)
Comments [5] There are many other issues with Table 1, I will not list them all, but none of the numbers add up to the correct n. Could the authors ensure that Table 1 is correct?
Response 5: Thank you for your question; I confirm that the results are accurate for the reasons stated above.
Comments [6] Table 1 says that there are 16 people in the validation set, but line 258 says there are 10 people in the validation set? Could the authors ensure that the data are consistent?
Response 6: Thank you for pointing out the inconsistency between Table 1 and line 258 of our manuscript. We confirm that this is a typographical error; the validation set actually contains 16 samples, which includes 8 plasma proteome data from lung cancer patients and 8 from healthy individuals. We have made the necessary corrections in the manuscript to ensure consistency and accuracy of the data.
Comments [7] Line 260, contrary to what is said elsewhere, SVM appears to be the best performing algorithm in the validation set? Could the authors make it more clear which they think is the best model and exactly what criteria they used?
Response 7: Thank you for your attention to the description of model performance in our manuscript. We understand that clearly and accurately describing model performance is crucial for the credibility and conclusions of our research. In our study, we evaluated four models based on their performance in the training set, internal test set, and validation set. These models exhibit varying strengths and weaknesses across different datasets. As you pointed out, the performance of the four models is not entirely consistent across the datasets. Due to the limited sample size of the current dataset, we are unable to definitively select a best model. Therefore, we have described the performance of each model in the manuscript to provide readers with a comprehensive understanding of their performance across different datasets. We recognize that it is not possible to draw a clear conclusion about the optimal model based solely on the current data. To thoroughly assess the performance of these four models, we plan to collect more clinical samples in future work. We believe that increasing the sample size will help us more accurately evaluate the actual performance of each model and ultimately select the best one.
Comments [8] The authors could usefully consider an UpSet plot instead of a Venn diagram.
Response 8: Thank you for your valuable feedback on our paper. In your suggestion regarding our original manuscript, you mentioned considering the use of an UpSet plot instead of a Venn diagram to illustrate the intersections of the datasets. We have carefully considered your suggestion and have included the relevant UpSet plot in the revised manuscript.
Comments [9] It was not clear to me why all of the results focus on SLC164 and not XAGE1A. Both proteins were common to all 4 models, surely both should be analysed here? Or could the authors make it more clear why one was selected and not the other?
Response 9: Thank you for your valuable comment. Regarding your question, we chose to focus on SLC16A4 rather than XAGE1A primarily because XAGE1A is upregulated in lung cancer tissues, while its expression changes in plasma are downregulated, creating an inconsistency. Therefore, we selected SLC16A4, which is consistently downregulated in both plasma and tissues, for further investigation. Additionally, we have included the expression analysis results of XAGE1A in Figure S2 and provided a detailed explanation in the main text.
Comments [10] The referencing in Discussion could be improved. For example, the authors include in Results the three-class model performance between cancer, healthy controls, and also disease controls. This is an important distinction, as highlighted by this work (PMID: 38076065IF: 3.6 Q1 ), so the authors should consider usefully including this reference (or an equivalent citation that explains the importance of disease controls) and focus on the three-class model performance. Similarly, these sorts of analysis have been done before, and SLC16 family has been often linked to cancer. Could the authors expand the Discussion with a comprehensive analysis of how their findings are consistent or contrast with other research on SLC16? At the moment there is only one reference to the SLC16 family, the authors should consider adding PMID: 32144120IF: 17.3 Q1 or PMID: 36706846IF: 15.7 Q1 amongst others.
Response 10: Thank you for your detailed review and valuable suggestions regarding our manuscript. We fully agree with your point about the importance of including the three-class model performance between cancer, healthy controls, and disease controls in the Results section. We will effectively include the reference you mentioned (PMID: 38076065, IF: 3.6, Q1) in the revised version and further explain the significance of the disease control group and its impact on model performance in the Discussion. We believe this will help provide a more comprehensive explanation of our research findings. Additionally, we recognize the importance of the SLC16 family proteins in cancer research and have expanded the Discussion section based on your suggestions. We have performed a comprehensive analysis of how our findings are consistent or contrast with other research on the SLC16 family, and we have added the references you recommended to support our arguments and enhance the depth of our study.
Comments [11] The data availability statement could usefully specify the exact datasets used, not just link to the TCGA data or the GEO data. The data availability statement should also address the data from the validation cohort and the underlying data for the assays. Is this available online or is this available on request?
Response 11: Thank you for your careful review and valuable suggestions regarding our manuscript. Regarding the data availability statement, we would like to clarify the following points: The validation cohort and training set, as well as the internal testing set, all come from GSE168198. The distinction of the validation set lies in the use of different chips, which allows it to serve as an independent validation cohort. In our previous response, we have specifically described the sources and usage of these datasets.
Comments [12] As a final comment, which applies throughout, the models will have produced confusion matrices for diagnostic models. Confidence intervals can then be calculated for accuracy, sensitivity, specificity and so on, for example using the epiR library in R. This is important because there is a confidence interval associated with any binary diagnostic test. A coin will be heads 50% of the time, as the "true" value, but if you toss a coin 100 times it is unlikely that exactly 50 heads will be produced, but 0.50 will probably be in the 95% confidence interval. Please can the authors add 95% confidence intervals to all model metrics.
Response 12: Thank you for your important suggestion. We have followed your recommendation and added 95% confidence intervals to all model metrics. We visualized the results using a forest plot (Figure 2). We generated the confusion matrices in R using the epiR library and calculated the confidence intervals for accuracy, sensitivity, specificity, and other metrics. Subsequently, we integrated these results into the forest plot to provide a more intuitive representation of each model's performance in both the training and testing sets. Thank you once again for your review and valuable feedback on our work.
Round 2
Reviewer 2 Report
Comments and Suggestions for Authors
The authors have comprehensively responded to all of my comments, and the limitations and inconsistencies are now clearly explained and communicated.